# Purified Gymnemic Acids from *Gymnema inodorum* Tea Inhibit 3T3-L1 Cell Differentiation into Adipocytes

**DOI:** 10.3390/nu12092851

**Published:** 2020-09-17

**Authors:** Papawee Saiki, Yasuhiro Kawano, Takayuki Ogi, Prapaipat Klungsupya, Thanchanok Muangman, Wimonsri Phantanaprates, Papitchaya Kongchinda, Nantaporn Pinnak, Koyomi Miyazaki

**Affiliations:** 1Cellular and Molecular Biotechnology Research Institute, National Institute of Advance Industrial Science and Technology (AIST), Tsukuba, Ibaraki 305-8566, Japan; y.kawano@aist.go.jp (Y.K.); k-miyazaki@aist.go.jp (K.M.); 2Department of Environment and Natural Resources, Okinawa Industrial Technology Center, Okinawa 904-2234, Japan; ogitkyuk@pref.okinawa.lg.jp; 3Research and Development Group for Bio-Industries, Thailand Institute of Scientific and Technological Research (TISTR), Techno Polis, Khlong Luang, Pathum Thani 12120, Thailand; prapaipat@tistr.or.th (P.K.); thanchanok@tistr.or.th (T.M.); Wimonsri@tistr.or.th (W.P.); papitchaya@tistr.or.th (P.K.); nantaporn_p@tistr.or.th (N.P.)

**Keywords:** *Gymnema inodorum*, adipogenesis, gymnemic acid, obesity

## Abstract

*Gymnema inodorum* (GI) is an indigenous medicinal plant and functional food in Thailand that has recently helped to reduce plasma glucose levels in healthy humans. It is renowned for the medicinal properties of gymnemic acid and its ability to suppress glucose absorption. However, the effects of gymnemic acids on adipogenesis that contribute to the accumulation of adipose tissues associated with obesity remain unknown. The present study aimed to determine the effects of gymnemic acids derived from GI tea on adipogenesis. We purified and identified GiA-7 and stephanosides C and B from GI tea that inhibited adipocyte differentiation in 3T3-L1 cells. These compounds also suppressed the expression of *peroxisome proliferator-activated receptor gamma* (*Pparγ*)-dependent genes, indicating that they inhibit lipid accumulation and the early stage of 3T3-L1 preadipocyte differentiation. Only GiA-7 induced the expression of *uncoupling protein 1* (*Ucp1*) and *pparγ coactivator 1 alpha* (*Pgc1α*), suggesting that GiA-7 induces mitochondrial activity and beige-like adipocytes. This is the first finding of stephanosides C and B in *Gymnema inodorum*. Our results suggested that GiA-7 and stephanosides C and B from GI tea could help to prevent obesity.

## 1. Introduction

*Gymnema sylvestre* is a species of the genus Gymnema that is popular in India for reducing glucose levels, suppressing glucose absorption and preventing type 2 diabetes [1,2,3,4,5]. *Gymnema inodorum* (GI) is a species of same genus that is indigenous to Thailand, particularly in the northern region, where it is widely consumed. The effects of GI on glucose absorption and blood glucose levels have recently been investigated [6,7,8]. We previously found that extracts of GI leaves decreased blood glucose in alloxan-induced diabetic rats [9] that comprise a popular model with which to study type 1 diabetes mellitus. Alloxan selectively destroys insulin production in beta cells, which consequently results in high blood glucose levels [10]. However, about 90% of patients with diabetes have type 2 diabetes mellitus (DM) which is induced by a lack of exercise and inappropriate eating habits [11]. However, obesity is the leading risk factor for type 2 DM, and it also greatly increases the risk of fatty liver disease, atherosclerosis, metabolic diseases, insulin resistance and hypertension [12,13]. Obesity is characterized at the cellular level as being differentiated from preadipocytes. White adipose tissue (WAT) is specialized to store excess energy as triglycerides composed of fatty acids. Inhibiting preadipocyte differentiation can prevent the initiation and progression of obesity [14,15].

The differentiation of 3T3-L1 fibroblast-like cells into adipocyte-like cells stimulated by insulin and synthetic glucocorticoids is a popular model of adipogenesis and lipid metabolism *in vitro* [16,17]. Therefore, we applied the inhibition of 3T3-L1 cell differentiation to screen gymnemic acid extracted from GI tea. Gymnemic acid is an oleanane-type triterpene glycoside [16,17] that can exist as a single entity or as a mixture of several related compounds [18,19]. The major saponin fraction in *Gymnema sylvestre* is a gymnemic acid that comprises a complex mixture of at least nine similar glycosides and aglycone derivatives [20]. Moreover, only four gymnemic acids have been identified in GI, which renders the purification and identification of gymnemic acids difficult. Furthermore, current knowledge about these compounds purified from GI is limited. We isolated and purified GiA-7, stephanoside C and stephanoside B from GI tea that inhibited 3T3-L1 cell differentiation. We also determined the expression of the *peroxisome proliferator-activated receptor gamma (Ppar*γ*), CCAAT/enhancer-binding protein alpha (Cebp*α*), cluster of differentiation 36 (Cd36), fatty acid synthase (Fasn), ppar*γ *coactivator 1 alpha (Pgc1α), lipin-1, adipose triglyceride lipase (Atgl), hormone-sensitive lipase (Hsl), sterol regulatory element-binding protein (Srebp)-1c, uncoupling protein 1 (Ucp1), glucose transporter type 4 (Glut4)* and *fatty acid binding protein 4 (Fabp4)* genes to explain the signaling of adipogenesis inhibition in 3T3-L1 preadipocytes.

## 2. Materials and Methods

### 2.1. Extraction, Isolation and Purification

Fresh GI leaves (Development of Herbs and Fruit Products Community Enterprise (Chiang Mai, Thailand)) were powdered, washed, dried and then steamed for 3 min. The leaves were dried at 60 °C for 2 h, stir-fried to complete dryness and then stored in darkness.

After extracting GI tea powder with 98% methanol for 24 h, the extract was mixed with hexane in a separatory funnel. The lower solution was collected, evaporated to dryness and then the residue was washed with chloroform and methanol (2:1) to remove fat components. The washed, evaporated residue dissolved in methanol (crude gymnemic acid) was eluted through a Sep-Pak tC_18_ cartridge (Waters Corporation, Milford, MA, USA) with a gradient of 10–100% methanol and ethanol. Six active compounds were purified from the 90% methanol fraction by high-performance liquid chromatography (HPLC) using a Model CCPD computer-controlled pump (Tosoh, Tokyo, Japan) equipped with a Capcell PAK C_18_ 5 µm, 20-mm inner diameter (i.d.), 250-mm column (Osaka Soda Co., Ltd., Osaka, Japan) and isocratic 80% methanol with 0.1% formic acid at a flow rate of 2.5 mL/min. The compounds were detected at 254 nm using a UV wavelength detector (JASCO International Co., Ltd., Tokyo, Japan).

### 2.2. Mass Spectrometry

Dried purified compounds were dissolved and diluted in dimethyl sulfoxide (Fujifilm Wako Pure Chemical Industries Ltd., Osaka, Japan) at 100 ppm. The accurate molecular formula was determined by Liquid Chromatography equipped with Quadrupole Time Of Flight Mass Spectrometry (LC/Q-TOF MS) using an Agilent 6530 Accurate-Mass Q-TOF LC/MS system (Agilent Technologies Inc., Santa Clara, CA, USA) equipped with an electrospray ionization (ESI) interface. Compounds were separated by reversed-phase liquid chromatography using a photodiode array detector and monitored at a wavelength ranging from 210 to 600 nm at a flow rate of 0.4 mL/min using an ACQUITY UPLC BEH C_18_ column (50 × 2.1 mm i.d. and 1.7 μm particle size (Waters Corp.) at 40 °C). The mobile phase consisted of a linear gradient of 0.1% formic acid:acetonitrile (1:1) to 0.1% formic acid:acetonitrile (1:19) over 3 min. The high-resolution mass spectra (HRMS) conditions were: positive ion mode; desolvation gas, N_2_; temperature 350 °C, pressure, 40 psig; flow rate, 8 L/min and capillary, fragmentary and skimmer voltages of 3500, 100 and 65 V, respectively [18].

### 2.3. Nuclear Magnetic Resonance (NMR) Spectroscopy

Dried compound 2 (10 mg) was exchanged into methanol-d4, 99.8 atom% D, containing 0.05% (*v*/*v*) Tetramethylsilane (TMS) (Cambridge Isotope Laboratories Inc., Andover, MA, USA). Dried compounds 5 and 6 (10 mg each) were exchanged into pyridine-d5, 99.5 atom% D (Cambridge Isotope Laboratories Inc.). Spectra were determined by one-dimensional (^1^H NMR, ^13^C NMR and dept-135) and two-dimensional COrrelated SpectroscopY (COSY), Heteronuclear Multiple Bond Correlation (HMBC) and Heteronuclear Multiple Quantum Coherence (HMQC) NMR using a Bruker 500 MHz NMR (Bruker Daltonics SPR, Hamburg, Germany).

### 2.4. Cell Culture

The mouse embryonic fibroblast cell line (3T3-L1 cell) was purchased from National Institutes of Biomedical Innovation, Health, and Nutrition (NIBIOHN), Osaka, Japan. These cells were cultured in low-glucose Dulbecco’s modified Eagle’s medium (DMEM) (Fujifilm Wako Pure Chemical Corp.) containing 10% heat-inactivated fetal bovine serum (FBS; Biowest, Tokyo, Japan) at 37 °C under a humidified 5% CO_2_ atmosphere.

### 2.5. The Antiadipocyte Differentiation Activity

We seeded 3T3-L1 cells (1 × 10^5^/mL in 200 µL) cultured as described above into collagen-coated 96-well plates in high-glucose DMEM (Fujifilm Wako Pure Chemical Corp.) under standard conditions for 24 h, then induced their differentiation into adipocytes using 10 µg/mL insulin, 0.5 mM 3-isobutyl-1-methylxanthine (IBMX) and 1 µM water-soluble dexamethasone (Sigma-Aldrich Corp., St. Louis, MO, USA). After 1 h, the 3T3-L1 cells were incubated with samples for 7–10 days.

Cell proliferation was determined using CellTiter 96^®^ AQueous One Solution Cell Proliferation Assays (Promega Corp., Madison, WI, USA), as described by the manufacturer. The absorbance of proliferating cells determined at 490 nm using an iMark^TM^ Microplate Reader (Bio-Rad Laboratories Inc., Hercules, CA, USA) was compared with that of untreated differentiated 3T3-L1 cells.

Intracellular lipid accumulation was determined using a Lipid Assay Kit (Cosmo Bio Co., Ltd., Tokyo, Japan), as described by the manufacturer. Differentiated 3T3-L1 cells were washed with phosphate-buffered saline (PBS), and fixed overnight with 4% formaldehyde at room temperature. The cells were then washed twice with distilled water, incubated with Oil red O at room temperature for 15 min and washed twice with distilled water. Oil red O extraction reagent was added into cells. Absorbance was read at 540 nm using the iMark^TM^ Microplate Reader. Absorption due to the intracellular lipid accumulation was determined and compared with that of control-differentiated 3T3-L1 cells.

### 2.6. Quantitative Real-Time PCR

We incubated 3T3-L1 cells (1 × 10^5^/mL; 1 mL) seeded into collagen-coated 12-well plates in high-glucose DMEM under standard conditions for 3 days, then induced the cells to differentiate into adipocytes using 10 µg/mL insulin, 0.5 mM IBMX and 1 µM water-soluble dexamethasone for 1 h. The 3T3-L1 cells were then incubated with purified GiA-7 and stephanosides C and B from GI tea (100 µM each) for 8 days. The expression of genes associated with adipogenesis was analyzed using quantitative real-time PCR. Total RNA was extracted from the cells using RNAiso plus. Single-stranded cDNA was generated using PrimeScript^TM^ RT Master Mix. Quantitative real-time PCR was conducted using a SYBR^®^ Premix Ex Taq™ II (Takara Bio. Inc., Otsu, Japan) and a LightCycler^TM^ (Roche Diagnostics, Mannheim, Germany). The sequences of all primers (Thermo Fisher Scientific Inc) are listed in Table 1 [19]. The PCR conditions were 95 °C for 10 s, followed by 45 cycles of 95 °C for 5 s, 58 °C for 10 s and at 72 °C for 10 s. The amount of target mRNA was normalized relative to the internal standard *36b4*.

### 2.7. Statistical Analysis

Data were statistically assessed by one-way analyses of variance (ANOVAs) with Dunnett tests using EZR software version 1.52 (Jichi Medical University, Saitama, Japan), which is graphical user interface for R (The R Foundation for Statistical Computing) based on R commander [20]. Values are indicated as means ± SD. Significant differences are shown as *p*-values.

## 3. Results and Discussion

We measured the ability of the crude 10–100% methanol and ethanol fractions of gymnemic acid to inhibit 3T3-L1 cell differentiation. We found that the 90% methanol fraction was the most powerful inhibitor (Figure 1). We then found that compounds 2, 3, 5 and 6 among the six compounds separated by HPLC from this fraction (Figure 2) significantly inhibited 3T3-L1 cell differentiation (Figure 3). Compound 5 was the most powerful inhibitor. The inhibition of 3T3-L1 cell differentiation by compound 5 was concentration-dependent. Compound 6 also strongly inhibited 3T3-L1 cell differentiation. However, the yield of HPLC fraction 3 was very low. Therefore, HPLC fractions No. 2, 5 and 6 were further purified, and their structures were identified by NMR and mass spectrometry. The ^13^C NMR chemical shifts of compounds 2, 5 and 6 were compared with published ^13^C NMR chemical shifts of GiA-7 [6], stephanoside C and stephanoside B [21], respectively, and are shown in Table 2, Table 3, Table 4, Table 5 and Table 6, respectively. These findings showed that compounds 2, 5 and 6 were GiA-7, stephanoside C and stephanoside B, respectively.

The molecular formulae of purified compounds 2, 5 and 6 were determined using Q-TOF LC/MS in the positive ion mode. The molecular formula of GiA-7 was C_44_H_65_NO_12_, according to the mass spectra (*m*/*z* 800.4580 (M + H)^+^, calcd. *m*/*z* 800.4582). Those of stephanoside C and stephanoside B were the same: C_52_H_79_NO_18_, calcd. 1006.5370. The accurate masses of stephanosides C and B were *m*/*z* 1006.5383 (M + H)^+^ and *m/z* 1006.5488 (M + H)^+^, respectively. The molecular formulae of stephanosides C and B are the same, but their sugar chains are d-thevetose and d-allomethylose, respectively. Appendix A shows the structures of compounds 2, 5 and 6. The NMR and mass spectrometry data confirmed that compounds 2, 5 and 6 are GiA-7, stephanoside C and stephanoside B, respectively.

We assessed the ability of purified 25, 50 and 100-µM GiA-7, stephanoside C and stephanoside B extracted from GI tea to inhibit 3T3-L1 cell differentiation. After 10 days, intercellular lipid accumulation and viable cells were determined. Each of GiA-7, stephanoside C and stephanoside B at 100 µM reduced intercellular lipid accumulation (Figure 4). Stephanoside C was the most effective inhibitor, which is the lowest concentration of significantly inhibited 3T3-L1 cell differentiation. Moreover, the inhibition of 3T3-L1 cell differentiation by GiA-7, stephanoside C and stephanoside B was concentration-dependent.

Several markers associated with adipogenesis control 3T3-L1 cell differentiation [22]. We assessed the expression of the *Pparγ*, *Cebpα*, *Fasn*, *Pgc1α, Cd36* and *Fabp4* genes that are associated with differentiation into adipocytes to determine the effects of GiA-7, stephanoside C and stephanoside B on adipogenesis. Figure 5 shows that GiA-7, stephanoside C and stephanoside B at 100 µM significantly suppressed the expression of *Pparγ*, *Cebpα*, *Fasn* and *Cd36*. Stephanoside B and GiA-7 significantly suppressed *Fabp4* expression. Stephanosides C and B also significantly suppressed *Pgc1α* expression. These results indicated that GiA-7, stephanoside C and stephanoside B inhibited the early stage of adipogenic differentiation by inhibiting of *Pparγ*-dependent mechanisms. Both Hsl and Atgl are phosphorylates upon appropriate physiological signaling to induce triacylglycerol (TG) lipolysis in adipocytes [23,24]. Figure 5 shows that stephanosides C, B and GiA-7 suppressed *Hsl* and *Atgl* gene expressions. These findings suggested that none of these compounds activated TG lipolysis. However, *Pparγ* directly regulates *Hsl* and *Atgl* gene expressions in adipocytes *in vitro* [25,26]. Our results suggested that these compounds downregulated *Hsl* and *Atgl* gene expressions by inhibiting *Pparγ* gene expression. Lipin-1 functions in lipid droplet biogenesis during adipocyte differentiation and generates diacylglycerol for lipid synthesis [27]. Lipin-1 is important for the process of TG accumulation during the early stage of adipogenesis. Lipin-1 is a key factor for adipocyte maturation and maintenance by regulating *Pparγ* and *Cebpα* [28]. *Lipin-1* expression is required to induce the transcription of adipogenic genes, including *Pparγ* and *Cebpα* [29,30]. Figure 5 shows that stephanosides C and B and GiA-7 significantly suppressed *lipin-1* expression. These findings suggest that these compounds inhibited *Pparγ* and *Cebpα* gene expressions by suppressing *lipin-1* gene expression. These observations confirm that GiA-7, stephanoside C and stephanoside B inhibited the early stage of adipogenesis and prevented TG accumulation. The transcriptional cofactor, Pgc1α, is important for mitochondrial biogenesis. The regulation of *Pgc1α* expression enhances mitochondrial biogenesis through *Srebp-1c* upregulation [31,32]. The present study found that only GiA-7 induced *Srebp-1c* and *Pgc1α*. Srebp-1c is also a key regulator of adipocytes and is involved in lipid metabolism [33,34]. These findings suggest that GiA-7 regulates mitochondrial biogenesis through the *Srebp-1c*-dependent upregulation of *Pgc1α*. GiA-7 also inhibits lipid accumulation in 3T3-L1 preadipocytes by downregulating adipogenic transcription factors and genes associated with lipid accumulation. Both *Pgc1α* and *Ucp1* are brown/beige cell-specific genes. Only GiA-7 induced the expression of *Pgc1α* and *Ucp1*. Beige adipocytes express low basal levels of *Ucp1*, whereas brown adipocytes constitutively express *Ucp1*. These findings suggest that GiA-7 inhibits the differentiation of white adipocytes and, also, induces beige-like adipocytes in 3T3-L1 mouse preadipocytes. Comprehensive profiles of gene expressions indicate that the characteristics of human brown and mouse beige adipocytes are compatible [33,34]. The activation of human brown adipocytes was recently examined as a possible novel therapeutic treatment for obesity [35]. Thus, GiA-7 might serve as a novel treatment for obesity in humans by inducing brown adipocytes.

Gymnemic acid extracted from the leaves of *Gymnema sylvestre* comprises a mixture of triterpene glycosides that can reduce glucose levels and inhibit glucose absorption [36,37,38]. The aqueous extract of *Gymnema sylvestre* induces insulin secretion in MIN6 cells [39]. One study found that GiA-7 from GI leaves inhibits glucose absorption in the isolated intestinal tract and suppresses blood glucose in rats [6]. However, we found here that GiA-7 purified from gymnemic acid extracted from GI tea inhibited 3T3-L1 cell differentiation into adipocytes. Stephanoside C and stephanoside B isolated from the stems of *Stephanotis lutchuensis var. japonica* and *Gongronema nepalense* have ant-malarial activity [21,40]. This is the first report of stephanoside C and stephanoside B isolated from *Gymnema inodorum* inhibiting 3T3-L1 cell differentiation into adipocytes. As mentioned before, obesity is characterized at the cellular level as being differentiated from preadipocytes. GiA-7, Stephanoside C and stephanoside B present in GI tea inhibited preadipocyte differentiation by suppressing the *Pparγ*-dependent mechanisms. These findings suggest that consuming GI tea could play a role in the prevention of obesity.

## 4. Conclusions

*Gymnema inodorum* tea has been widely applied in Thailand to control high blood glucose. Here, we screened the ability of gymnemic acids extracted from GI tea to inhibit 3T3-L1 cell differentiation into adipocytes. We isolated and purified GiA-7, stephanoside C and stephanoside B from GI tea using column chromatography and C18 HPLC, respectively, then confirmed them using NMR and mass spectrometry. All three compounds inhibited 3T3-L1 cell differentiation into adipocytes. Moreover, we determined that these compounds inhibited the early stage of adipogenesis by suppressing the *Lipin-1, Pparγ, Cebp*α*, Fasn, Cd36* and *Fabp4* genes that are associated with adipogenesis. However, only GiA-7 induced *Ucp1* and *Pgc1α*, suggesting that GiA-7 enhances mitochondrial activity and beige-like adipocytes among 3T3-L1 preadipocytes. Our findings suggest that the GiA-7, stephanoside C and stephanoside B from GI tea could help to prevent obesity.

## 5. Patents

Papawee Saiki and Yasuhiro Kawano, the methods of inhibiting fat synthesis, fat synthesis inhibitors and food and drink for suppressing fat synthesis, JP patent 2019-218481.

## Figures and Tables

**Figure 1 nutrients-12-02851-f001:**
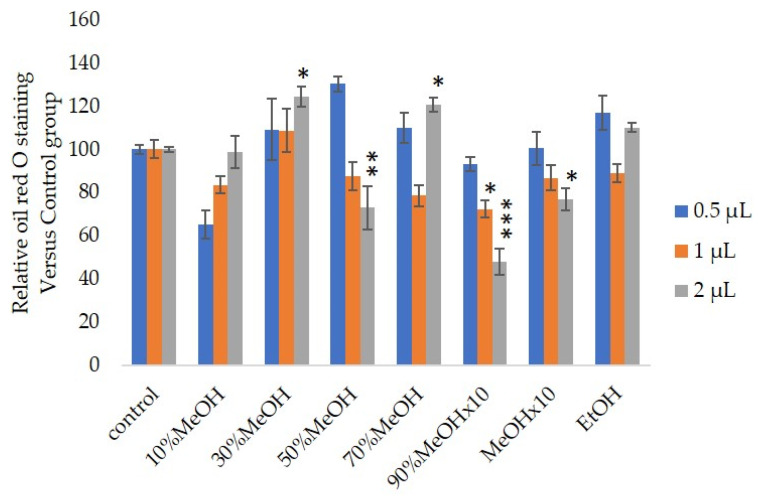
Effects of Sep-PaktC18 fractions on 3T3-L1 cell differentiation. We assessed the abilities of 10%MeOH, 30%MeOH, 50%MeOH, 70%MeOH and EtOH fractions at the concentration of 10 mg/mL in ethanol and 90%MeOH and MeOH fractions at the concentration of 1 mg/mL in ethanol to inhibit 3T3-L1 cell differentiation. Values are shown as means ± SD (*n* = 4). * *p* < 0.05, ** *p* < 0.01 and *** *p* < 0.001 vs. control (ANOVA with post hoc Dunnett tests).

**Figure 2 nutrients-12-02851-f002:**
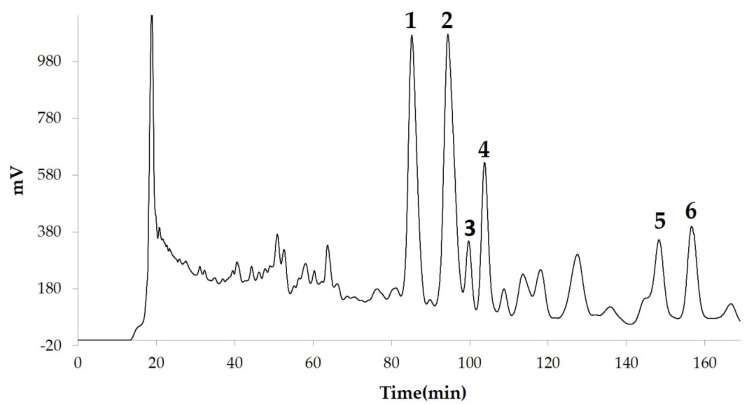
Compounds separated by high-performance liquid chromatography (HPLC) from 90% methanol fraction.

**Figure 3 nutrients-12-02851-f003:**
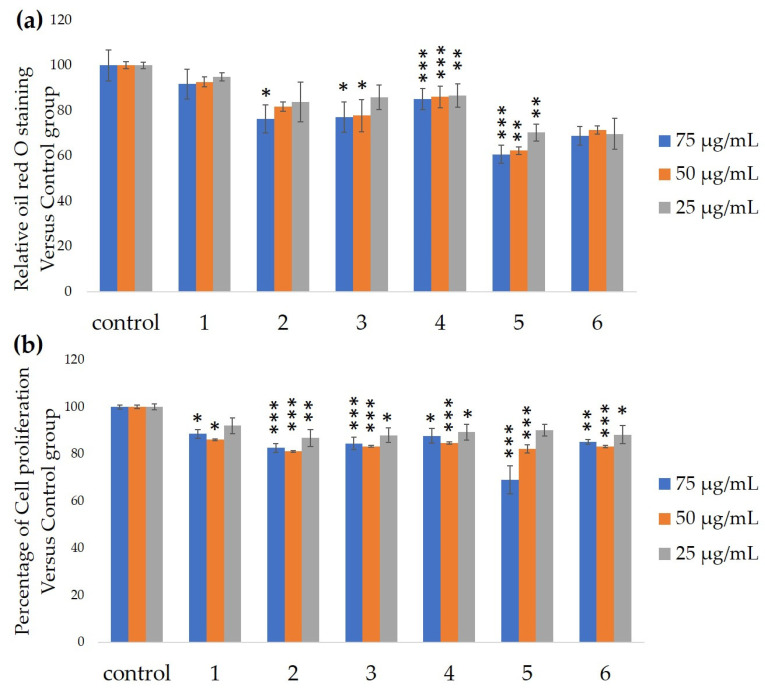
Ability of HPLC fractions to inhibit 3T3-L1 cell differentiation. (**a**) Inhibition of adipogenesis. (**b**) Cell proliferation. Values are shown as means ± SD (*n* = 4). * *p* < 0.05, ** *p* < 0.01 and *** *p* < 0.001 vs. control (ANOVA and post hoc Dunnett tests).

**Figure 4 nutrients-12-02851-f004:**
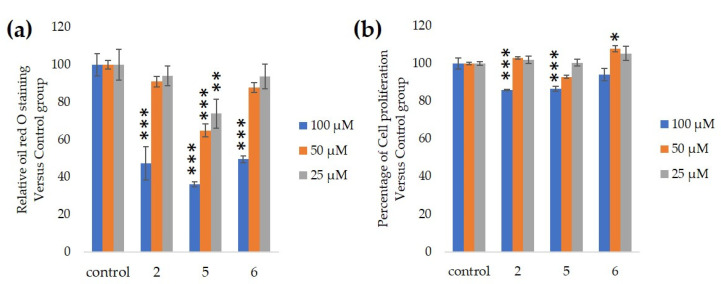
Effects of compounds 2 (GiA-7), 5 (stephanoside C) and 6 (stephanoside B) on 3T3-L1 cell differentiation. (**a**) Inhibition of adipogenesis. (**b**) Cell proliferation. Values are shown as means ± SD (*n* = 4). * *p* < 0.05, ** *p* < 0.01 and *** *p* < 0.001 vs. control (ANOVA and post hoc Dunnett tests).

**Figure 5 nutrients-12-02851-f005:**
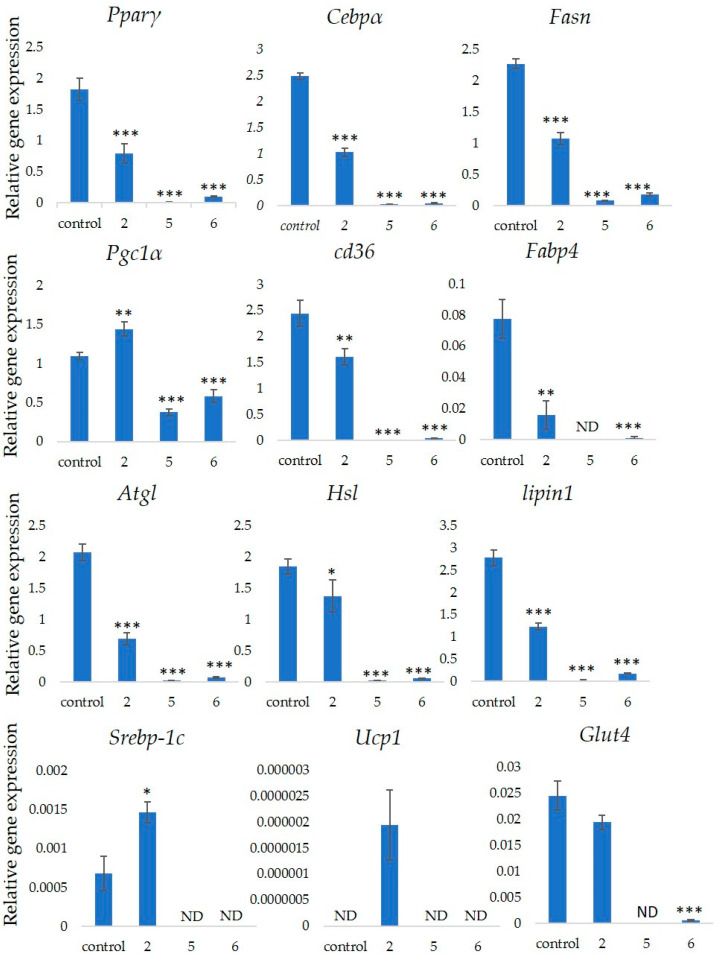
Effects of stephanosides C and B and GiA-7 extracted from *Gymnema inodorum* (GI) tea on gene expressions at the initial stage of 3T3-L1 cell differentiation into adipocytes. The differentiation of 3T3-L1 cells was induced, and the cells were incubated with 100 µM GiA-7, stephanoside C and stephanoside B for 8 days; then, the gene expressions were measured. Values are shown as means ± SD (*n* = 4). * *p* < 0.05, ** *p* < 0.01 and *** *p* < 0.001 vs. control (ANOVA with post hoc Dunnett tests. ND, not detected).

**Table 1 nutrients-12-02851-t001:** Primer sequences for real-time reverse transcription (RT)-PCR.

Target Gene	Direction	Primer Sequence (5′–3′)
*Pparγ*	Forward	AACTCTGGGAGATTCTCCTGTTGA
	Reverse	TGGTAATTTCTTGTGAAGTGCTCATA
*Fasn*	Forward	GGAGGTGGTGATAGCCGGTAT
	Reverse	TGGGTAATCCATAGAGCCCAG
*Cebpα*	Forward	AAGAAGTCGGTGGACAAGAACAG
	Reverse	GTTGCGTTGTTTGGCTTTATCTC
*Pgc1α*	Forward	GTAGGCCCAGGTACGACAGC
	Reverse	GCTCTTTGCGGTATTCATCCC
*Lipin-1*	Forward	CCATAGAGATGAGCTCGGAT
	Reverse	AACTGGGATACGATGCTGACT
*Atgl*	Forward	CTTGAGCAGCTAGAACAATG
	Reverse	GGACACCTCAATAATGTTGGC
*Hsl*	Forward	GCTGGAGGAGTGTTTTTTTGC
	Reverse	AGTTGAACCAAGCAGGTCACA
*Srebp-1c*	Forward	ATCGGCGCGGAAGCTGTCGGGGTAGCGTC
	Reverse	ACTGTCTTGGTTGTTGATGAGCTGGAGCAT
*Glut4*	Forward	CTGTCGCTGGTTTCTCCAAC
	Reverse	CAGGAGGACGGCAAATAGAA
*Ucp1*	Forward	GGCAACAAGAGCTGACAGTAAAT
	Reverse	GGCCCTTGTAAACAACAAAATAC
*Fabp4*	Forward	CCGCAGACGACAGGA
	Reverse	CTCATGCCCTTTCATAAACT
*36b4*	Forward	CTTCATTGTGGGAGCAGACA
	Reverse	TCTCCAGAGCTGGGTTGTTC

**Table 2 nutrients-12-02851-t002:** ^13^C nuclear magnetic resonance (NMR) chemical shifts of GiA-7 and compound **2** (δ: ppm).

C-No.	Carbon Type	GiA-7	Compound 2
1	—CH_2_—	39.7	39.7
2	—CH_2_—	26.2	26.4
3	>CH—O—	82.3	82.9
4	>C<	43.9	44.0
5	>CH—	48.1	48.1
6	—CH_2_—	18.8	18.9
7	—CH_2_—	33.2	33.2
8	>C<	41.2	41.3
9	>CH—	48.2	48.2
10	>C<	37.5	37.5
11	—CH_2_—	24.8	24.8
12	—CH=	124.9	125.0
13	>C=	142.8	142.8
14	>C<	43.9	44.0
15	—CH_2_—	37.0	37.0
16	>CH—O—	66.8	66.8
17	>C<	46.5	46.5
18	>CH—	44.9	44.9
19	—CH_2_—	47.1	47.1
20	>C<	33.0	33.1
21	—CH_2_—	39.9	39.9
22	>CH—	74.3	74.3
23	—CH_2_—O—	64.8	64.7
24	—CH_3_	13.4	13.4
25	—CH_3_	16.7	16.7
26	—CH_3_	17.5	17.5
27	—CH_3_	28.0	28.0
28	—CH_2_—O—	61.2	61.1
29	—CH_3_	33.5	33.5
30	—CH_3_	25.6	25.6
O-NMAt			
N1	>C=O	169.6	169.7
N2	>C=	112.1	112.1
N3	>C=	153.0	153.0
N4	—CH=	111.9	112.0
N5	—CH=	135.6	135.6
N6	—CH=	115.3	115.4
N7	—CH=	133.0	133.0
N8	—CH_3_	29.7	29.7
β-glu			
1’	—O—CH—O—	105.3	105.7
2’	>CH—O—	75.0	75.2
3’	>CH—O—	78.0	78.0
4’	>CH—O—	73.5	73.5
5’	>CH—O—	76.6	78.0
6’	—COO—		

**Table 3 nutrients-12-02851-t003:** The ^13^C NMR chemical shifts of stephanoside C and compound **5** (δ: ppm).

C-No.	Carbon Type	Stephanoside C	Compound 5
1	—CH_2_—	38.9	38.9
2	—CH_2_—	30.0	30.0
3	>CH—O—	77.8	77.7
4	—CH_2_—	39.3	39.4
5	>C=	139.3	139.4
6	—CH=	119.5	119.5
7	—CH_2_—	35.0	35.0
8	>C<	74.4	74.4
9	>CH—	44.1	44.2
10	>C<	37.3	37.4
11	—CH_2_—	25.7	25.7
12	>CH—	74.7	74.6
13	>C<	57.0	57.0
14	>C<	89.0	89.0
15	—CH_2_—	33.8	33.8
16	—CH_2_—	34.0	34.0
17	>C<	87.7	87.7
18	—CH_3_	11.4	11.4
19	—CH_3_	18.1	18.1
20	>CH—O—	75.0	75.0
21	—CH_3_	15.6	15.7
12-O-Acetyl moiety			
A1	—COO—	171.5	171.4
A2	—CH_3_	22.1	22.1
20-O-*N*-Methylanthraniloyl moiety			
N1	—COO—	111.0	111.1
N2	>C=	152.7	152.7
N3	>C=	111.6	111.6
N4	CH=	135.1	135.2
N5	—CH=	114.8	114.8
N6	—CH=	132.7	132.7
N7	—CH=	168.3	168.3
N8	—CH_3_	29.7	29.6

**Table 4 nutrients-12-02851-t004:** The ^13^C NMR chemical shifts of sugar chains of stephanoside C and compound **5** (δ in ppm).

C-No.	Carbon Type	Stephanoside C	Compound 5
d-Cymarose			
1′	—O—CH—O—	96.5	96.5
2′	—CH_2_—	37.3	37.4
3′	>CH—O—	78.0	78.0
4′	>CH—O—	83.5	83.5
5′	>CH—O—	69.1	69.1
6′	—CH_3_	18.7	18.7
O-Me	—O—CH_3_	59.0	59.0
d-Olenadrose			
1′′	—O—CH—O—	102.2	102.1
2′′	—CH_2_—	37.7	37.8
3′′	>CH—O—	79.3	79.3
4′′	>CH—O—	83.2	83.2
5′′	>CH—O—	72.1	72.2
6′′	—CH_3_	19.0	19.0
O-Me	—O—CH_3_	57.4	57.4
d-Thevetose			
1′′′	―O—CH—O—	104.2	104.3
2′′′	>CH—O—	75.3	75.4
3′′′	>CH—O—	88.2	88.3
4′′′	>CH—O—	76.1	76.1
5′′′	>CH—O—	72.9	73.0
6′′′	—CH_3_	18.8	18.8
O-Me	—O—CH_3_	61.1	61.1

**Table 5 nutrients-12-02851-t005:** The ^13^C NMR chemical shifts of stephanoside B and compound **6** (δ: ppm).

C-No.	Carbon Type	Stephanoside B	Compound 6
1	—CH_2_—	38.8	38.8
2	—CH_2_—	30.0	29.9
3	>CH—O—	77.7	77.6
4	—CH_2_—	39.3	39.2
5	>C=	139.3	139.2
6	—CH=	119.4	119.4
7	—CH_2_—	34.9	34.9
8	>C<	74.3	74.3
9	>CH—	44.1	44.0
10	>C<	37.3	37.2
11	—CH_2_—	25.6	25.6
12	>CH—	74.6	74.6
13	>C<	56.9	56.9
14	>C<	88.9	88.9
15	—CH_2_—	33.8	33.7
16	—CH_2_—	33.9	33.9
17	>C<	87.6	87.6
18	—CH_3_	11.3	11.3
19	—CH_3_	18.1	18.0
20	>CH—O—	74.9	74.9
21	—CH_3_	15.6	15.6
12-O-Acetyl moiety			
A1	—COO—	171.3	171.3
A2	—CH_3_	22.0	22.1
20-O-*N*-Methylanthraniloyl moiety			
N1	—COO—	111.0	111.0
N2	>C=	152.6	152.6
N3	>C=	111.5	111.5
N4	—CH=	135.1	135.1
N5	—CH=	114.7	114.7
N6	—CH=	132.6	132.6
N7	—CH=	168.2	168.2
N8	—CH_2_	29.6	29.5

**Table 6 nutrients-12-02851-t006:** The ^13^C NMR chemical shifts of sugar chains of stephanoside B and compound **6** (δ: ppm).

C-No.	Carbon Type	Stephanoside B	Compound 6
d-Cymarose			
1′	—O—CH—O—	96.4	96.4
2′	—CH_2_—	37.3	37.2
3′	>CH—O—	77.9	77.9
4′	>CH—O—	83.5	83.5
5′	>CH—O—	69.0	68.9
6′	—CH_3_	18.7	18.7
O—Me	—O—CH_3_	58.9	58.8
d-Olenadrose			
1′′	—O—CH—O—	101.9	101.9
2′′	—CH_2_—	37.5	37.6
3′′	>CH—O—	79.3	79.2
4′′	>CH—O—	82.8	82.9
5′′	>CH—O—	72.0	72.0
6′′	—CH_3_	19.0	18.9
O-Me	—O—CH_3_	57.2	57.2
d-Allomethylose			
1‴	—O—CH—O—	102.2	102.1
2‴	>CH—O—	73.2	73.3
3‴	>CH—O—	84.0	84.1
4‴	>CH—O—	74.6	74.5
5‴	>CH—O—	71.0	71.0
6‴	—CH_3_	18.7	18.7
O-Me	—O—CH_3_	62.1	62.1

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
