# Peer review of "Purified Gymnemic Acids from Gymnema inodorum Tea Inhibit 3T3-L1 Cell Differentiation into Adipocytes"

_nutrients, 2020, doi:10.3390/nu12092851_

Round 1

Reviewer 1 Report

 I reviewed the Manuscript ID nutrients-921784: Purified gymnemic acids from Gymnema inodorum tea inhibit 3T3-L1 cell differentiation into adipocytes.

The argument is very interesting.

The purpose of the authors in this work was to determine the effects of gymnemic acids derived from GI tea on adipogenesis. The authors purified and identified β-D-glucopyranosiduronic acid (GiA-7) and stephanosides C and D from GI tea that showed to inhibit adipocyte  differentiation in 3T3-L1 cells. These compounds also showed to suppress the expression of Lipin-1, PPARγ, Cebpa, Fasn, Cd36 and Fabp4, indicating that they inhibit lipid accumulation and the early stage of  3T3-L1 preadipocyte differentiation. Moreover,  GiA-7 displayed to induce the expression of Ucp1 and Pgc1a,  suggesting that GiA-7 induces mitochondrial activity and beige-like adipocytes. The authors suggest, based on the obtained results, that GiA-7,  stephanosides C and D from GI tea could help to prevent obesity.

Overall, this work was well designed and performed with proper scientific methodology.  

However, the article is penalized by the fact that Results and Discussion section is not written fully clearly. In some cases, the results are poorly described and discussed, in other cases some concepts are expressed in a redundant way.

I suggest to the authors to discuss the results about the adipogenesis inhibition activity, and cellular differentiation inhibition  of different amounts of  GiA-7,  stephanosides C and D from GI tea also in terms of inhibition or not of cellular proliferation (cellular viability), figures 3 and 4.

Other comments;

Results and Discussion section:

Lines 158-161:  I don't understand what the square brackets contain…

Lines 184-188: check the font size

Lines 190-191: check the caption of figure 4

Lines 220-221: “ …”the characteristics of human brown, and mouse beige adipocytes are compatible” What do the authors mean by this sentence?

Some typing errors in the text

Author Response

Response to Reviewer 1 Comments

Point 1:

However, the article is penalized by the fact that Results and Discussion section is not written fully clearly. In some cases, the results are poorly described and discussed, in other cases some concepts are expressed in a redundant way.

I suggest to the authors to discuss the results about the adipogenesis inhibition activity, and cellular differentiation inhibition  of different amounts of  GiA-7,  stephanosides C and D from GI tea also in terms of inhibition or not of cellular proliferation (cellular viability), figures 3 and 4.

Response 1: We discussed more as advice in Line 150-152 and 191-192

Point 2:

Lines 158-161:  I don't understand what the square brackets contain…

Response 2: They are our mistakes. We revised already.

Point 3:

Lines 184-188: check the font size

Response 3: We checked and revised.

Point 4:

Lines 190-191: check the caption of figure 4

Response 4: We checked and revised.

Point 4:

Lines 220-221: “ …”the characteristics of human brown, and mouse beige adipocytes are compatible” What do the authors mean by this sentence?

Response 4: We mean that we found that GiA-7 induced beige adipocytes in 3T3-L1 cells (mouse cells). But characteristics of human brown, and mouse beige adipocytes are compatible. So, we think we can use GiA-7 for obesity treatment in human. However, we revised to easily understand.

Point 5:

Some typing errors in the text

Response 5: We checked.

Reviewer 2 Report

This is an interesting study by Saiki et al that investigates the effects of gymnemic acid on adipogenesis. The authors have extracted three compounds from GI leaves, purified and characterized them using Mass Spectrometry and NMR. Although it looks straight forward but it’s not as simple and easy and therefore deserves appreciation. In the present study, Saiki et al have tested the effects of these compounds on the differentiation ability of 3T3-L1 mouse preadipocytes into adipocytes. They elegantly show that all three compounds block differentiation of 3T3-L1 preadipocytes into fully differentiated adipocytes via different mechanisms. This study, therefore, provides a nice molecular link into how consumption of GI leaves (and or three of the ingradients found in these leaves) could reduce adipogenesis and help reduce the development of obesity.

This study would be of great interest to the broad readership of this journal. However, I have couple of minor comments (as follows) that I would like to be taken care.

Comments:

  • Line 101--103 Cell Culture section: Authors make the following statement

“The mouse embryonic fibroblast cell line, 3T3-L1(National Institutes of Biomedical Innovation, Health, and Nutrition [NIBIOHN), Osaka, Japan), has the characteristics of pancreatic beta cells, including insulin secretion in response to glucose.”

I am not aware of any study which claims that 3T3-L1 cells can produce insulin in response to glucose. I would like to read this particular article. Therefore, could authors please cite that article?

  • Line 121 Oil red O extraction was added into cells.

I think authors want to say that Oil red O stain was added to the cells.

  • Line 201-202 Both Hsl and Atgl are phosphorylates that increase triacylglycerol (TG) lipolysis in adipocytes

I think authors want to say that HSL and ATGL are phosphorylated upon appropriate physiological signaling to induce TG lipolysis in adipocytes.

Authors should also discuss the connection between PPARγ, HSL and ATGL while discussing figure 5. HSL and ATGL both are PPARγ regulated genes (See the references below PMID: 16269451 & 17848638). Since treatment with all three compounds downregulates PPARγ expression (Fig 5), you would expect HSL and ATGL expression to be downregulated as well. Authors results in Fig 5 show exactly that. It needs to be discussed and the molecular connection be established.

Similarly, the connection between Lipin1 and PPARγ is also missing. Lipin1 regulates PPARγ transcriptional activity. It’s likely that these compounds downregulate Lipin1 which then downregulates PPARγ and as a result HSL & ATGL are also downregulated.

Furthermore, authors have not highlighted their findings (specifically the molecular link, Figure 5 is the novelty of the whole study and that is why I am recommending this study for publication). Authors should discuss all this as I have mentioned and cite the provided references accordingly.

The rationale (Why shall people consume GI leaves? What is the molecular connection between consuming these leaves and controlling obesity? How GI leaves reduce the development of obesity?) is missing from the discussion. The ability of GI leaves to reduces glucose absorption in the intestine and adiposity as a result has been reported by multiple studies for nearly two decades. Therefore, authors should highlight the molecular aspect of this study (that’s the meat) while discussing figure 5 or in the conclusion section.

References:

  1. PPARgamma regulates adipose triglyceride lipase in adipocytes in vitro and in vivo. Erin E Kershaw1, Michael Schupp, Hong-Ping Guan, Noah P Gardner, Mitchell A Lazar, Jeffrey S Flier    PMID: 17848638   PMCID: PMC2819189    DOI: 1152/ajpendo.00122.2007
  2. Peroxisome Proliferator-Activated Receptor-γ Transcriptionally Up-Regulates Hormone-Sensitive Lipase via the Involvement of Specificity Protein-1 Tuo Deng, Song Shan, Ping-Ping Li, Zhu-Fang Shen, Xian-Ping Lu, Jing Cheng, Zhi-Qiang Ning Endocrinology, Volume 147, Issue 2, 1 February 2006, Pages 875–884, https://doi.org/10.1210/en.2005-0623
  3. Lipin1 regulates PPARγtranscriptional activity HeeEun Kim,* Eunju Bae,* Deok-yoon Jeong,* Min-Ji Kim,* Won-ji Jin,* Sahng-wook Park,* Gil-Soo Han, George M. Carman, Eunjin Koh,*,1 and Kyung-Sup Kim*,1         PMCID: PMC3690191                 PMID: 23627357

  • Data in figure 4a shows that stephanoside C is most effective in inhibiting adipogenesis among the three compounds tested. It would be interesting to see the additive and or synergistic effect of stephanoside C in combination with GiA-7 and or stephanoside B. It is likely that stephanoside C could inhibit adipogenesis more effectively at a much lower concentration in combination with other compounds. Therefore, I suggest this simple experiment that can be finished in a relatively short period of time.

There are several other interesting questions (few mentioned below) that can be addressed but don’t have to be the part of this manuscript. 

  1. Does GiA-7 induce mitochondrial biogenesis or enhance mitochondrial activity?
  2. Although, all three compounds inhibit adipogenic differentiation but Glut4 expression in GiA-7 treated cells is as good as control cells while it’s dramatically downregulated by the other two compounds. What is the status of glucose uptake in GiA-7 treated cells versus the other two compound treated cells?
  3. What’s the status of insulin signaling and the intermediates in the cells treated with these three compounds?
